# Investigating Mineral Accumulation and Seed Vigor Potential in Bottle Gourd (*Lagenaria siceraria*) through Crossbreeding Timing

**DOI:** 10.3390/plants12233998

**Published:** 2023-11-28

**Authors:** Anurag Malik, Virender Singh Mor, Himani Punia, D. S. Duhan, Axay Bhuker, Jayanti Tokas, Mohamed A. El-Sheikh, Tariq Shah

**Affiliations:** 1Department of Seed Science & Technology, College of Agriculture, CCS Haryana Agricultural University, Hisar 125004, Haryana, India; 2Division of Research and Innovation, Uttaranchal University, Dehradun 248007, Uttarakhand, India; 3Department of Sciences, Chandigarh School of Business, Chandigarh Group of Colleges, Jhanjeri, Mohali 140307, Punjab, India; 4Department of Biochemistry, College of Basic Sciences & Humanities, CCS Haryana Agricultural University, Hisar 125004, Haryana, India; 5Department of Vegetable Science, College of Agriculture, CCS Haryana Agricultural University, Hisar 125004, Haryana, India; 6Botany and Microbiology Department, College of Science, King Saud University, Riyadh 11451, Saudi Arabia; melsheikh@ksu.edu.sa; 7Helmholtz Center for Environmental Research, Theodor-Leiserstr 4, 06120 Halle, Germany

**Keywords:** bottle gourd, crossing periods, crude protein, quality, temperature, nutrition

## Abstract

Bottle gourd (*Lagenaria siceraria*) is a well-known cucurbit with an active functional ingredient. A two-year field experiment was carried out at the Research Farm of Seed Science and Technology, CCS HAU, Hisar, in a randomized block design during the Kharif season (Kharif is one of the two major cropping seasons in India and other South Asian countries, heavily reliant on monsoon rains with the other being Rabi) and the summer season. Five different crossing periods (CP), viz. CP1, CP2, CP3, CP4, and CP5, were considered to illustrate the effects of agro-climatic conditions on the quality and biochemical components of two bottle gourd parental lines and one hybrid, HBGH-35. The average mean temperature for the Kharif season in 2017 was 31.7 °C, and for the summer season, it was 40.1 °C. Flowers were tagged weekly from the start of the crossing period until the end and harvested separately at different times. The fruits harvested from different crossing periods under different environmental conditions influenced the bottle gourd’s qualitative and biochemical traits and showed significant variations among the five crossing period environments. A positive significance and correlation were observed between weather variables and different biochemical characteristics. Henceforth, the CP_4_ crossing period at a temperature of 31.7 °C retained high-quality seed development, which may be essential in enhancing agricultural productivity and the national economy.

## 1. Introduction

Bottle gourd (*Lagenaria siceraria* (Molina) Standl) is an edible, useful, and medicinal vegetable crop belonging to the Cucurbitaceae family, valued for its culinary, therapeutic, and useful properties [1]. This cross-pollinated vegetable exhibits varying degrees of pollination, ranging from 94% to 99%. The degree of cross-pollination depends on several factors, such as the flowering time, temperature, wind velocity and direction, planting design, insect population, and genotypic nature, ultimately determining the kernel quality [2,3,4]. The reproductive phase in cucurbits, initiated approximately six to seven weeks after planting, is critical in deciding the seed quality due to the source–sink relationship dynamics [5,6]. As a result of the source–sink strength relationship, a variation in seed lots can be observed [7]. Being indeterminate, bottle gourd vines can withstand adverse weather conditions like continuous dry and overcast weather such as seed germination, emergence, seedling establishment, leaf canopy development, flowering, and seed maturation.

Seed quality is paramount in agricultural production and food security, especially during the growing uncertainty caused by climate change [8,9,10] and abiotic factors [11,12]. Hybrids, carrying the combined genetic makeup of plants, are pivotal in agricultural biotechnology and crop improvement, with high-quality seeds being in constant demand to ensure productive plantations. Farmers constantly need high-quality sources to ensure efficient and effective plantations; thus, companies must sell high-quality seeds to maintain their competitive position in markets [13]. Numerous abiotic factors, predominantly high and low temperatures, low light, moisture surpluses [9,13,14], and deficits impose limitations on the growth, development, and yield of different crops [15,16]. Cool nights are favorable during the fruiting period, but extreme or low temperatures may result in delayed growth and aborted fruiting sites. Temperature significantly affects leaf expansion, phenology, internode elongation, assimilation partitioning, and biomass production [17]. The variations in protein and lipid contents were attributed to several environmental factors [16,17,18]. The timing of seed development is crucial, as it directly impacts the accumulation of essential minerals and nutrients necessary for robust seedling growth [18,19].

The mineral content of seeds correlates with seedling quality. During seed development, there is a critical need to accumulate minerals and essential nutrients, particularly in the early growth stage [19,20]. The minerals collected during seed ontogeny are the primary factors influencing seedling development; growth would be delayed without these components [20]. To understand the mechanism behind the source–sink relationship and improve the nutritional profiles of bottle gourd crops, it becomes crucial to examine the interactions between macro- and micronutrients during various stages of cross-breeding [21]. To maintain the crop’s nutritional profile, seeds contain enough metabolic reserves to facilitate seedling establishment during seed development. Carbohydrates, lipids, and proteins interact synergistically to create an optimal environment for seed germination and seedling development. To develop bottle gourd genotypes with improved nutritional and phytochemical compositions, micronutrient and macronutrient profiling of the germplasm is essential [22]. Before delving into the study of gene activity influencing the inheritance of nutritional and antinutritional traits in both parent and offspring plants, it is imperative to identify the optimal cross-breeding periods for the desired genotypes.

Quality is pivotal in crop production, necessitating specific attributes and functionality during the developmental phase [4]. The efficiency of hybrid seed production depends on multiple factors, such as selecting an appropriate agro-climate location, a suitable season, improved floral synchronization for enhanced seed setting through suitable crossing periods, supplementary pollination techniques, etc. [4]. Ref. [4] found that bottle gourd exhibits bi-hemispheric adaptability and distribution, thriving in tropical and temperate regions. It flourishes in areas with annual rainfall ranging from 400 to 1500 mm, with a preference for moderate soil moisture levels over excessive ones for optimal harvests. According to Grubben and Dento [23], bottle gourd grows well at temperatures of 25–35 °C. The optimum germination temperature is between 20 and 25 °C. Temperatures below 15 °C and above 35 °C reduce the germination rate [24]. Agronomic practices that promote the production of more female flowers than male flowers could increase yields; however, Haque et al. [25] observed fewer seed sets due to the reduction in pollen. Therefore, it is essential to determine the optimum ratio of male and female flowers to optimize the fruit and seed set.

Nevertheless, the effects of varying environmental conditions on bottle gourd seed properties remain inadequately understood, underscoring the importance of studying mineral accumulation during different fruit-setting periods at various temperatures. Thus, the present study aims to gain a deeper understanding and clarify the impact of pollination vulnerability and the outcomes resulting from varying cross-breeding timings on the biochemical composition of bottle gourd.

## 2. Materials and Methods

### 2.1. Plant Material and Experimental Setup

The research experiment was conducted at the experimental farm of Seed Science & Technology (29.1416° N, 75.7112° E, with an average elevation of 215 m (705 ft) above mean sea level) during the Kharif (2017) and summer season (2018) in a randomized block design. The soil type of the experimental field is sandy loam. The concentrations of organic matter percent, total nitrogen, available phosphate, and rapidly available phosphate in the uppermost 20–30 cm soil were 0.49%, 182 kg/ha, 18 kg/ha, and 285 kg/ha, respectively. The climate is semi-arid, with freezing winters and hot, dry, desiccating winds during the summer.

The seeds of bottle gourd parental lines, G-2 (Male Line) and Pusa Naveen (Female Line), were procured from the Department of Vegetable Science to produce Hybrid HBGH-35. For the production of hybrid HBGH-35, seeds of the bottle gourd parental lines G-2 (male line) and Pusa Naveen (female line) were obtained from the Department of Vegetable Science. HBGH-35 is the first hybrid developed by the CCS HAU, Hisar, India, using the G2 male line from Gujarat regions, and the female lines used was PUSA Naveen developed by IARI Pusa, both of which are popularly grown in the northern part of India. G2 is an early flowering line, and Pusa Naveen is a high-yielding line. These lines were selected to compare the potential of HBGH-35 concerning parental lines.

Being an indeterminate crop, flowering in bottle gourd continues for around two months. Sowing occurred on 17 July, and flowering began in both parents in the first week of September. There were ten vines per plot with a size of 6 × 3 m, and the vine-to-vine spacing was 60 cm. Every experiment was carried out in three biological replicates and five technical triplicates. All methods followed the protocols based on the relevant guidelines and regulations. A standard recommended package of practices (POP) was followed throughout the experiment.

### 2.2. Pollination and Emasculation

The flowers (male and female) were emasculated one day before opening, and they were wrapped in a thin layer of cotton. The following day, pollen was collected from the male flower and dusted on the female flower, which was then wrapped with a thin layer of cotton until the fruit formation started.

Monoecious bottle gourds (*Lagenaria siceraria*) bear male and female flowers. Key inferior ovary variations between male and female flowers were as follows:

The male flower has a long, slender peduncle and one pollen-producing stamen. Male flowers lack or have a primitive ovary. Male bottle-gourd flowers lack or have a primitive ovary. Female flowers have a unique ovary at the base that will produce fruit if fertilized. The bulbous ovary is at the floral center. Well-developed female flowers have ovules in their ovaries. A functioning ovary is essential for female flowers. Bottle gourds and other monoecious plants generate more male flowers than female flowers. This may be an adaptive technique to ensure pollination with enough pollen. Environmental conditions and plant health affect the flower output and ratio. To determine flower sex and optimize pollination, growers watch the flowering pattern. If natural pollination is insufficient, farmers may hand-pollinate to increase the fruit set.

After bagging, we waited for one week and observed female flower ovary growth. The increased size of the ovary confirmed the successful crossing after it was tagged using different color threads, as shown in the picture. The pollinated female flower was tagged for further identification.

The emasculation and pollination work were carried out in both parental lines with the help of needles, scalpers, and forceps. The two glass cups, cotton, muslin cloth, plastic bag, camel brush, and scissors were used to collect pollen from male flowers and dust it on the stigmatic surface of emasculated female flowers for hybrid seed production. Manual emasculation and dusting were continued throughout the crossing period. Subsequently, we pinched off all the floral buds emerging beyond 15 days after the crossing period to ensure better growth and production of crossed fruits (Figure 1). The harvesting dates of different crossing periods are listed in Table 1.

At the start of the crossing period in hybrid and parental line seed production, the male flower was chosen in the parental line, bagged with paper bags the previous evening, pinched off the following day, and kept carefully in a glass container. The fruits were harvested from the hybrid and parental lines at maturity when the color turned from green to brown. Seeds were extracted manually by slicing the ripened fruits into small pieces, and seeds were separated by squeezing the fruit pulp. The extracted seeds were cleaned with fresh water and dried to lower the moisture content.

### 2.3. Soil Profile

The concentrations of organic matter percent, total nitrogen, available phosphate, and rapidly available phosphate in the uppermost 20–30 cm soil were 0.49%, 182 kg/ha, 18 kg/ha, and 285 kg/ha, respectively.

### 2.4. Agro-Meteorological Data

The harvested seeds of five crossing periods of all genotypes were studied for various biochemical parameters under varying weather conditions. Agro-meteorological data were recorded at the agro-meteorological observatory during the seed development period of both seasons and converted to weekly data, presented in Table 2 and Figure 2.

### 2.5. Biochemical Analysis

The seeds were extracted from fruits set in each crossing period manually. After extraction, seeds are dried in the shade to lower the moisture content to 8% and be stored for further biochemical analysis. The moisture content of the seed was measured using the hot oven method. In this method, the pre-weighed aluminum boxes and seed material are placed in an oven maintained at 103 °C. Seeds are dried at this temperature for 1 hr. The relative humidity of the ambient air in the laboratory must be less than 70 percent when the moisture determination is carried out.

The moisture content as a percentage by weight (fresh weight basis) is calculated to one decimal place by using the following formulae:
$$\begin{array}{ll} \mathrm{The}\mathrm{percentage}\mathrm{of}\mathrm{seed}\mathrm{moisture}\mathrm{content}\mathrm{(mc)}= & \mathrm{M2}-\mathrm{M3}\times 100\\ & \mathrm{M2}-\mathrm{M1} \end{array}$$

M1 = Weighing container weight with cover in gramsM2 = Weighing container weight with cover and seeds before dryingM3 = After-drying weight of the weighing container with cover and seeds

Total soluble solids (°Brix) were determined by using a hand refractometer. The seeds’ ash, moisture content, and crude fiber content were determined using the recommended method [26]. The estimated total nitrogen (N) and crude protein were calculated by multiplying 4.27 by the N concentration [26]. The lipid content was determined using the Soxhlet method. Total carbohydrate was estimated as the method described by Dubois et al. [27]. Vitamin C was measured using the titration method of reduction of 2,6-dichlorophenol-indophenol (DCPIP) dye described [28]. The energy value was calculated by multiplying the mean values of crude protein, lipid content, and total carbohydrate by factors of 4, 9, and 4, respectively.

### 2.6. Determination of Macro and Micro-Elements

The mature seeds (50) were dried at constant temperature, then ground to fine powder. Random sampling was performed to ensure that the samples represented the entire mature dried bottle gourd seed population. The measurements of macro- and microelements in dried seeds were carried out using energy dispersive X-ray (EDX) analysis and atomic absorption spectrometry (AAS). Three biological and five technical replicates from each sample were used for analysis.

### 2.7. Energy Dispersive X-ray Analysis (EDX)

Three seeds from each selection were cryo-fractured in liquid nitrogen and split into halves using a scalpel blade. Seeds were mounted on stubs and secured using double-sided insulating carbon tape. Seed mineral composition was determined separately for the embryo, cotyledon, and seed coat regions.

### 2.8. Atomic Absorption Spectrometry

The contents of the elements were determined in the seed (2 to 3 fruits per vine) materials after incineration in a muffle furnace at 660 °C. The resultant ash content was solubilized on porcelain crucibles using a 10 mL acid mixture (HNO_3_ and HCl mixed in a 1:3 ratio) [29] using an atomic absorption spectrophotometer (Alpha 4 Model). A rapid sequential absorption spectrometer was used to evaluate potassium using flame atomic emission spectroscopy (FAES) (Varian AA280FS). Flame atomic absorption spectroscopy was used to determine the concentrations of Ca, Mg, Zn, Cu, Fe, and Mn (FAAS). Table 3 summarizes the components evaluated at each wavelength, recovery level, and relative standard deviation (RSD). Using a Varian Alpha UV-VIS spectrometer at 400 nm, the amount of phosphorus was measured (Spectronic Unicam, Berlin, Germany) [30].

### 2.9. Measurement of Germination Axis

#### 2.9.1. Standard Germination

One hundred seeds were taken in four replications from each treatment and placed uniformly between two sufficiently moist germination papers. They were then rolled and placed in a germinator, where the temperature and relative humidity were kept at 25.1 °C and 95.1%, respectively. The final counts were made on the 14th day of the germination test for normal seedlings and expressed in percentages as per the ISTA protocol [31].

The germination percentage was worked out using the following formula:$$Germination\left(\%\right)=\frac{Number\;of\;normal\;seedlings}{The\;total\;number\;of\;seeds\;plante}\times 100$$

#### 2.9.2. Biomass Accumulation

The plants were uprooted and thoroughly washed with distilled water to eliminate the soil. To remove surface moisture, a paper towel was used. An electronic balance (Presica 105A) was used to measure fresh weights. Weighed fresh samples were heated in an air oven at 70 °C for 24 h and then dried to a constant weight at 35 °C. The plant dry weight was equal to the shoot and root dry weights. The root-shoot ratio was calculated according to Agren and Ingestad [32].
$$Root-shoot\;ratio=\frac{Dry\;weight\;for\;roots}{Dry\;weight\;for\;the\;top\;of\;the\;plant}$$

#### 2.9.3. Root and Shoot Length

A two-factorial experiment was replicated three times to measure the germinant axis and provide experimental units. Sand particles were removed from seedlings developed through sand soil by uprooting and washing them. The lengths of the roots and shoots were measured using computer-assisted image analysis. A tabloid scanner (Hewlett Packard Scanjet 4c/t; Palo Alto, CA, USA) was used to capture digital photos at a scanning resolution of 250 dpi. The lengths of the roots and shoots were measured using image software analysis.

### 2.10. Statistical Analysis

The data were presented as mean SD, and (three replicates)-way ANOVA was conducted to check the significance of the main effects (cultivars and temperature seasons) and their interaction on growth indices, followed by a post hoc comparison (Tuckey’s test) at a 5% level (*p* < 0.05). Statistical analysis was performed using SPSS v25.0 software (SPSS for Windows, Chicago, IL, USA). The correlations between phytochemicals and weather variables were statistically evaluated and indicated by Pearson’s coefficient indexes using two-tailed bivariate correlates analysis; *p* < 0.05 was considered statistically significant.

## 3. Results

### 3.1. Proximate Composition

Seed production in bottle gourds is complicated because seed development occurs within moist tender bottle gourd fruits for several weeks, and the seeds are frequently kept wet before seed harvest. The moisture content varied significantly among the bottle gourd cultivars under different growing seasons (*p* < 0.05) (Table 4). The male parental line (G2-line) had the highest moisture content (12.7%), followed by the female one with 12.8% at 31.7 °C. Hybrid HBGH-35 retained its minimum moisture content (12.4%) at 31.7 °C during the Kharif season, while at 40 °C (summer season), it decreased by 12.2, 11.6, and 11.8%, respectively, during the fourth week of the crossing period (CP-4). Variation in seed moisture is due to the flesh of fruits from CP-1, which get more time to dry out than those from CP-5. Secondly, summers have more hot winds/temperatures, so the fruits dry out faster than in the Kharif season. During the summer, there was a sharp increase in moisture content, leading to poor seed quality, and a sharp increase in the moisture content, leading to poor seed quality and deterioration. The ash content indicated no significant variation (*p* > 0.05) among different crossing periods but within different growing seasons (Table 4). Hybrid HBGH-35 had the highest ash content (5.8%), followed by Pusa Naveen (5.2%), while the minimum was in the male line (4.2%) in the Kharif season (31.7 °C) during the fourth week of the crossing period. The results presented in Table 4 depict crude fiber content in bottle gourd samples that varied significantly (*p* < 0.05), with Hybrid HBGH-35 having the highest value (48.3%), followed by Pusa Naveen (37.2%), while the G2-line seed fiber (30.3%) was the least at 31.7 °C. With the increase in temperature, the crude protein declined and affected the overall seed fiber quality.

### 3.2. Biochemical Constituents

The variation in TSS observed in fruits of exotic bottle gourd cultivars was significant (Table 5). The highest value of TSS (8.2 °Brix) was recorded in Hybrid HBGH-35 at 31.7 °C, while the values declined to 7.9 at 40 °C in CP-4. The minimum value of TSS (4.1 to 4.7 °Brix) was recorded in the male line at 31.7 °C and 40 °C, respectively. The parental lines had TSS ranging from 6.2 to 7.6 °Brix (♀) and 6.0 to 6.5 °Brix (♂), respectively, during the fourth week of the crossing period. All the cultivars had TSS values with some variation due to temperature fluctuations. As illustrated in Table 5, crude protein varied significantly (*p* < 0.05) among different growing seasons and crossing periods. The order of concentration is Hybrid HBGH-35 > Pusa Naveen > G2-line.

### 3.3. Total Carbohydrates and Ascorbic Acid

The carbohydrate content in Hybrid HBGH-35 retained its maximum value (22.1%), followed by Pusa Naveen (19.2%), while the G2-line seed fiber (15.5%) had the lowest value (Table 6). Temperature significantly affects sugar accumulation, thus, the overall energy changes. At 31.7 °C, the seeds accumulated a high carbohydrate content (8.9–22.1%) during the fourth week of the crossing period, while at 40 °C, the concentration decreased (8.1–20.3%) significantly. The observed difference among the crossing periods indicates that the distribution of carbohydrates in the seed tends to concentrate more at lower temperatures (31.7 °C). The observed trend was the same for the energy values expressed as kcal/100 g (Table 6) of the seeds harvested from different growing seasons. It could be attributed to the high crude protein and lipid content in the hybrid.

The data presented in Table 6 for vitamin C content in bottle gourd cultivars exhibited significant differences under different environmental conditions. Hybrid HBGH-35 had the highest vitamin C content (16.5 mg/100 g) followed by Pusa Naveen (14.6 mg/100 g), while the G2-line had a minimum (12.6 mg/100 g) during the fourth week of the crossing period at 31.7 °C. The fruits harvested during the summer (40 °C) accumulated lower concentrations (6.8–16.0%) of vitamin C in all cultivars. The hybrid HBGH-35 represented adaptability to the agro-climatic conditions of the study site.

### 3.4. Analysis of Energy Dispersive X-ray

An elemental composition comparable with the different minerals observed in the scanning electron microscope (SEM) was revealed via the energy dispersive x-ray (EDX) analysis of bottle gourd parental and hybrid lines. The key components that were found were largely consistent. It is clear from Table 7 that P, K, and Mg are abundant in each of the regions under investigation, with the embryonic region having the highest concentration. In this investigation, we showed that the seed coat had more calcium than the embryonic and cotyledon sections. The results suggest that Ca(I) forms Ca-pectin complexes with the carboxyl of pectin molecules inside the cell wall. Zn, Cu, and Mn are abundant in the embryonic area, and they are also found in large concentrations in the cotyledon tissue of bottle gourds and their seed coatings.

### 3.5. Macroelements

The P, K, and Mg mineral compositions of seeds from different parental lines of bottle gourds did not differ significantly (*p* > 0.05) (Table 7). The K, P, and Mg ratio in the seed coat area of the bottle gourd lines was relatively lower than the embryonic region composition. The hybrid HBGH35 and Pusa Naveen had higher mean values of P, K, and Mg than the G2 line.

### 3.6. Micro-Nutrients Analysis

Between the parental lines, the calcium levels varied significantly (*p* < 0.05). Hybrid HBGH-35 has a seed coat region with a calcium concentration of 2.14% and a G2 embryonic area with a calcium level of 0.45% (Table 8). HBGH-35 had the highest zinc level in the embryo area (1.43%), whereas G2 had the lowest proportion in the seed coat region (0.08%). Regarding copper, the embryonic area of HBGH-35 recorded the greatest percentage (0.39%), whereas the seed coat region of G2 recorded the lowest rate (0.03%). Significant variations were seen in the manganese percentage, with HBGH-35 having the most significant amount in the embryo area (0.31%) and G2 having the lowest amount in the seed coat region (0.02%).

### 3.7. Seed Macro- and Microelement Composition

The mineral content in seeds, particularly essential macro- and microelements like nitrogen, phosphorus, potassium, calcium, and magnesium, can influence the availability of nutrients for seed germination and early seedling growth. The imbalanced mineral content can also have negative effects on seed vigor. The results of the mineral contents present in bottle gourd seeds are shown in Table 9 and Table 10. Potassium (K) is the most abundant element in bottle gourd (Table 9). All the cultivars showed a significant variation (*p* < 0.05) in their potassium content, with the highest amount found in hybrid HBGH-35 (3015.2 mg/100 g) at the fourth week of the crossing period. In comparison, the minimum was found in the male line (2054.2 mg/100 g). At 31.7 °C, the concentration of sodium also varied significantly (*p* < 0.05). Among the bottle gourd lines, the fourth week of the crossing period had the highest potassium accumulation (1042.6–1412.2 mg/100 g). At 40 °C, these elements were reduced significantly (1365.3–1011.3 mg/100 g) (Table 9). The level of Ca in bottle gourd cultivars is generally low and shows no significant variations (*p* > 0.05) among the different growing seasons (Table 10). The lowest concentration of Ca was found in the male line (1.3 mg/100 g), followed by the female line (1.6 mg/100 g) and the hybrid (1.89 mg/100 g). While the P content was high in hybrids with a value of 1250.3 mg/100 g at the fourth week of the crossing period at 31.7 °C, there were no significant differences (*p* > 0.05). The P content of bottle gourd seeds was generally higher than in conventional seeds and nuts (11–350 mg/100 g) [10,33,34]. The magnesium (Mg) displayed significant variations (*p* < 0.05) between the analyzed samples, with the hybrid (570.6 mg/100 g) having the highest value, followed by Pusa Naveen (485.3 mg/100 g) seeds, and then the G2 line (449.6 mg/100 g) at 31.7 °C.

The hybrid (570.6 mg/100 g) had the highest value (15.2 mg/100 g) followed by Pusa Naveen (12.5 mg/100 g) seeds, then the G2 line (10.4 mg/100 g) at 31.7 °C (Table 9). Iron (Fe) plays a crucial role in the body’s hemoglobin synthesis and oxygen-carrying capacity. Among the samples analyzed, hybrid HBGH-35 (65.3 mg/100 g) accumulated a higher amount of Fe, followed by Pusa Naveen (51.2 mg/100 g) at 31.7 °C (Table 9). There is a significant difference in the Fe content in the two growing seasons, i.e., 19.6–65.3 mg/100 g to 15.1–55.3 mg/100 g at 31.7 °C and 40 °C, respectively. At 31.7 °C, hybrids retained their maximum Zn (0.20 mg/100 g) and Mn (26.4 mg/100 g) contents during the fourth week of the crossing period, while at 40 °C, the levels decreased by 0.18 mg/100 g and 24.3 mg/100 g, respectively, in all cultivars. The differences in the accumulation of minerals between cultivars might be due to their genetic makeup and their comparative responses to variations in agro-climatic conditions.

### 3.8. Correlation Analysis between Weather Variables and Biochemical Components

The correlation analysis of biochemical constituents with weather variables showed a significant variation among the five different crossing periods (Table 11). The protein content correlated significantly with minimum temperature and morning and evening relative humidity. Crude fiber (r = 0.956 **), TSS (r = 0.990 **), and total carbohydrate content (r = 0.813 *) had a highly positive correlation with minimum temperature. Vitamin C accumulation (r = 0.967 **) positively and significantly correlated with minimum temperature. With the temperature rise, the moisture content also increased, as shown by a positive correlation (r = 0.827 *) with maximum temperature, thus justifying the present study results. A graphical representation illustrating the relationship between the traits analyzed in the present study was obtained through a multivariate analysis showing PCA based on the Cos^2^ value. The Cos^2^ of the variables indicated that the moisture, CP, CF, lipids, vitamin C, and others traits in the positive direction and the fats and TCC in the negative direction were the major contributing traits (Figure 3).

**Table 10 plants-12-03998-t010:** Mineral composition of hybrid bottle gourd under summer season.

Season	Summer Season (2018–19)				
Crossing Periods	CP-1	CP-2	CP-3	CP-4	CP-5
Hybrid HBGH-35					
Sodium	1011.3 ± 1.3 ^a^	1069.3 ± 2.1 ^b^	1302.1 ± 2.0 ^d^	1365.3 ± 1.8 ^e^	1223.5 ± 1.9 ^c^
Potassium	2001.3 ± 2.1 ^a^	2086.3 ± 2.6 ^b^	2436.2 ± 3.2 ^d^	2865.2 ± 3.6 ^e^	231.75.1 ± 4.1 ^c^
Calcium	1.48 ± 0.21 ^e^	2.17 ± 0.23 ^d^	3.44 ± 0.32	3.65 ± 0.23	3.02 ± 0.21
Phosphorus	923.4 ± 3.1 ^a^	1001.3 ± 4.2 ^b^	1214.2 ± 2.6 ^d^	1235.6 ± 3.3 ^e^	1141.0 ± 2.4 ^c^
Magnesium	275.3 ± 4.3 ^a^	325.6 ± 3.6 ^b^	514.3 ± 3.2 ^d^	540.2 ± 3.9 ^e^	451.3 ± 2.2 ^c^
Iron	21.6 ± 1.3 ^e^	28.3 ± 1.9 ^d^	47.6 ± 3.2 ^b^	55.3 ± 4.3 ^a^	40.2 ± 2.9 ^c^
Copper	8.3 ± 0.12 ^e^	10.6 ± 0.12 ^d^	12.6 ± 0.16 ^b^	14.2 ± 0.19 ^a^	11.0 ± 0.13 ^c^
Zinc	0.07 ± 0.06 ^a^	0.09 ± 0.02 ^b^	0.14 ± 0.03 ^d^	0.18 ± 0.06 ^e^	0.11 ± 0.04 ^c^
Manganese	13.5 ± 0.15 ^e^	18.6 ± 0.12 ^d^	21.3 ± 0.16 ^b^	24.3 ± 0.12 ^a^	18.6 ± 0.16 ^c^
Pusa Naveen (♀)					
Sodium	946.2 ± 1.5 ^a^	1021.3 ± 1.6 ^b^	1264.2 ± 2.3 ^d^	1289.3 ± 2.1 ^e^	1186.6 ± 1.7 ^c^
Potassium	1856.3 ± 3.6 ^a^	1956.3 ± 4.5 ^b^	2365.3 ± 6.5 ^d^	2689.6 ± 3.2 ^e^	2236.5 ± 2.9 ^c^
Calcium	1.3 ± 0.23 ^e^	1.6 ± 0.12 ^d^	2.8 ± 0.13 ^b^	3.2 ± 0.23 ^a^	2.2 ± 0.12 ^c^
Phosphorus	865.3 ± 3.6 ^a^	984.3 ± 3.2 ^b^	1149.6 ± 2.3 ^d^	1132.6 ± 2.6 ^e^	1065.8 ± 3.6 ^c^
Magnesium	232.4 ± 5.2 ^a^	284.6 ± 3.6 ^b^	451.2 ± 2.3 ^d^	487.6 ± 4.3 ^e^	412.0 ± 3.6 ^c^
Iron	18.9 ± 1.9 ^e^	21.6 ± 1.7 ^d^	40.6 ± 2.2 ^b^	38.6 ± 2.6 ^a^	28.6 ± 2.1 ^c^
Copper	7.6 ± 0.15 ^e^	8.2 ± 0.14 ^d^	9.1 ± 0.21 ^b^	10.6 ± 0.12 ^a^	9.3 ± 0.08 ^c^
Zinc	0.04 ± 0.06 ^a^	0.06 ± 0.09 ^b^	0.11 ± 0.07 ^d^	0.15 ± 0.06 ^e^	0.09 ± 0.04 ^c^
Manganese	10.2 ± 0.03 ^e^	13.2 ± 0.06 ^d^	17.4 ± 0.10 ^b^	20.7 ± 0.13 ^a^	15.6 ± 0.10 ^c^
G2-line (♂)					
Sodium	895.3 ± 1.3 ^a^	983.2 ± 1.1 ^b^	1232.6 ± 2.6 ^d^	1254.2 ± 2.1 ^e^	1124.3 ± 2.0 ^c^
Potassium	1745.2 ± 2.1 ^a^	1846.5 ± 2.1 ^b^	2246.2 ± 3.5 ^d^	2564.3 ± 4.1 ^e^	2146.3 ± 2.6 ^c^
Calcium	1.0 ± 0.16 ^e^	1.3 ± 0.19 ^d^	2.2 ± 0.22 ^b^	3.0 ± 0.23 ^a^	1.9 ± 0.19 ^c^
Phosphorus	832.6 ± 4.3 ^a^	947.2 ± 6.3 ^b^	1084.3 ± 5.4 ^d^	1064.3 ± 3.2 ^e^	1011.3 ± 4.2 ^c^
Magnesium	384.3 ± 4.6 ^a^	220.6 ± 3.8 ^b^	402.6 ± 4.9 ^d^	412.5 ± 4.6 ^e^	375.6 ± 3.8 ^c^
Iron	15.1 ± 1.8 ^e^	18.7 ± 2.1 ^d^	33.4 ± 2.4 ^b^	31.6 ± 1.9 ^a^	23.4 ± 1.7 ^c^
Copper	6.2 ± 0.06 ^e^	7.4 ± 0.10 ^d^	8.3 ± 0.09 ^b^	8.9 ± 0.13 ^a^	8.5 ± 0.15 ^c^
Zinc	0.03 ± 0.01 ^a^	0.04 ± 0.02 ^b^	0.08 ± 0.03 ^d^	0.11 ± 0.09 ^e^	0.07 ± 0.04 ^c^
Manganese	8.9 ± 0.02 ^e^	10.2 ± 0.04 ^d^	13.2 ± 0.04 ^b^	17.6 ± 0.12 ^a^	11.6 ± 0.10 ^c^

^a–e^ Values with different superscripts are significantly different (*p* < 0.05); CP-1: 4–10 September 2017 and 6–22 April 2018; CP-2: 11–17 September 2017 and 23–29 April 2018; CP-3: 18–24 September 2017 and 30 April–6 May 2018; CP-4: 25 September–1 October 2017 and 7–13 May 2018; CP-5: 2–8 October 2017 and 14–20 May 2018; CP: crossing period.

**Table 11 plants-12-03998-t011:** Pearson’s correlation coefficient analysis between weather variables and biochemical constituents.

	T_max_	T_min_	RH_m_	RH_e_	Moisture	Lipid	CP	CF	TSS	TCC	Vit. C
T_max_	1	0.793	0.521	0.696	0.827 *	0.784 *	0.424	0.457	0.491	0.470	0.501
T_min_		1	0.208	0.810 *	0.410	−0.405	0.816 *	0.956 **	0.990 **	0.813 *	0.967 **
RH_m_			1	0.736	0.961 **	0.950 **	0.960 **	0.959 **	0.924 **	0.932 **	0.907 **
RH_e_				1	0.842 *	0.844 *	0.861 *	0.879 *	0.883 *	0.879 *	0.875 *
Moisture					1	0.923 **	0.993 **	0.968 **	0.925 **	0.921 **	0.913 **
Lipid						1	0.923 **	0.988 **	0.994 **	0.997 **	0.986 **
CP							1	0.964 **	0.919 **	0.923 **	0.900 **
CF								1	0.990 **	0.989 **	0.983 **
TSS									1	0.998 **	0.997 **
TCC										1	0.991 **
Vit. C											1

* Correlation is significant at the 0.05 level (2-tailed); ** correlation is significant at the 0.01 level (2-tailed); T_max_: maximum temperature; T_min_: minimum temperature; RH_m_: morning relative humidity; RH_e_: evening relative humidity; CP: crude protein; CF: crude fiber; TSS: total soluble solids; TCC: total carbohydrate content; Vit. C: vitamin C.

### 3.9. Seed Vigor

#### 3.9.1. Germination Indices

The germination was observed to be higher (95.84%) in the seeds harvested from the Kharif crop as compared to the summer crop (94.11%) (Table 12). Among the three genotypes studied, HBGH-35 had the highest standard germination rate (95.60%), followed by Pusa Naveen (94.93%), and G2 had the lowest (94.40%). Among the crossing periods, the lowest germination percentage (94.22%) was recorded in the seeds produced from crossing period CP1. A significant increase in germination percentage was observed from CP1 up to the CP4 crossing period, after which a significant decline was recorded in CP5. The maximum germination was recorded in the seeds of the CP4 treatment (95.94%), followed by the CP3 (95.17%).

#### 3.9.2. Growth Characteristics

The HBGH-35 and G2 parental lines’ hybrid root length, shoot length, root dry mass, and shoot dry mass were all substantially different (*p* < 0.01) from one another (Figure 4). At CP4, HBGH-35 and G2 had the greatest and lowest RL values, respectively (53.4 mm and 27 mm) (Figure 4a). At CP1 and CP3, HBGH-35 obtained the highest SL readings of 32 mm and 39.05 mm, respectively. At both sampling intervals, G2 had the lowest SL (Figure 4b). At CP4, HBGH-35 and G2 had the greatest (0.52 mg plant-1) and lowest (0.24 mg plant^−1^) root biomass measurements, respectively. RDM values were considerably elevated at CP4, with HBGH-35 recording the lowest biomass at 0.78 mg plant-1 and G2 recording the greatest biomass at 2.15 mg plant^−1^ (Figure 4c). After CP1 and CP4, respectively, the shoot dry mass varied from 0.34 to 0.7 mg plant-1 and from 2.71 to 5.20 mg plant^−1^, considerably varying between treatments. The SDM for HBGH-35 was the greatest, while the SDM for G2 was the lowest (Figure 4d).

## 4. Discussion

The estimation of the proximate composition in bottle gourd seeds is essential to determine the source–sink relationship. The relatively low moisture content in bottle gourd is advantageous since the high moisture content is associated with increased bacterial action during storage and is a crucial determinant of seed quality [35]. The results of this study are very similar to those of [36], which stated that the moisture content is essential in determining the seed quality. The effect of the moisture content during seed production in bottle gourds is a much more complicated process because the seed development occurs within the moist fruits for several weeks. Seeds are often held in a wet state before their final harvest. The ash content represents the index of mineral elements present in the seed. This indicates that the whole seed may have a higher mineral content [37] and be of good quality. The ash content in whole seeds is generally higher than the 3.7% reported as the ash content of the calabash seed [37,38]. Hassan et al. [37] observed that ash content represents an index of high minerals that improved the overall performance of seed quality. Dietary fiber promotes the food absorption and bowel movement through the small intestine [39]. Bottle gourd is a rich source of dietary fibers that expand food inside the colon wall, easing the passage of waste and making it an effective anti-constipation agent. The crude fiber content of the bottle gourd seed is comparable with that of calabash (3.2%) and watermelon (2.46%) [40].

Total soluble solids (TSS) measure the sugar content of seeds and play a significant role during seed production. The seeds harvested in the fourth week of the crossing period with a temperature of 31.7 °C had a higher protein content (8.2–19.1%), while with the temperature rise (40 °C), the protein content decreased (6.2–17.6%) significantly due to the inactivation of the enzymes involved in protein synthesis [41]. The protein forms the principal component of embryonic development; thus, this could be why the higher protein concentration in Hybrid HBGH-35 (19.1%) at CP-4 represents high-quality seeds [42]. Similar results were found [43] in soybean and in groundnuts [44]. Lipids represent the energy content required during seed germination. A similar trend was observed for the lipid content (Table 5). All the cultivars showed significant variations (*p* < 0.05) among them. However, the values are lower than the 41–46.8% reported in the same Lagenaria species [45]. Carbohydrates are a rich source of energy [10,46]. The results conform well with those observed in bottle gourd [37], watermelon [47], oil bean seed [48], pumpkin seed [49], and African walnut [50], which confirmed that seeds possessing higher carbohydrates have a high germination rate and seed emergence. The results are well documented by several researchers [51,52,53,54,55], who reported that temperature plays a crucial role in the accumulation of vitamin C. A higher vitamin C accumulation indicates a higher antioxidant potential of the seed [51].

Results for EDX are semi-qualitative estimations based on conventional analysis and theoretical intensity adjustments [56]. It has been reported that phytin globoids are prevalent in the periplasm of embryonic cells in legume seeds, and also that P, Mg, and K accumulate in the globoids. Calcium was scarcely found in the embryonic tissues of bottle gourd seeds. Similar observations were previously reported [57,58]. In the seed embryonic region, Zn, Cu, and Mn percentages associated with the P percentage suggested a potential relationship with phytate. Iwai et al. [59] reported that specific distribution patterns of minerals in seeds depend upon the cell type. However, this report was based on a single spot analysis, which involves analyzing areas’ electron beam size. Phytate synthesis in seeds deposits phosphates and cations for a sufficient period during germination [55]. According to the results of the present study, seeds may be a significant source of nutrients for catering to the requirements of seedling growth [60]. A high seed P content improves plant establishment and increases the dry matter accumulation. This results from a faster initial root growth, which gives seedlings earlier access to growth-limiting resources (water and mineral elements). According to Grant et al. [61], seed P reserves sustain a maximal growth of seedlings for several weeks after germination until the plant has three or more leaves and a substantial root system. Indole acetic acid (IAA)-induced cell wall weakening and solute aggregation inside the cell to establish internal osmotic potential are essential for cell expansion [62]. K+ buildup lowers the water potential and maintains the pH of growing cells. Since a stoichiometric K+ inflow electrochemically counterbalances the IAA-stimulated H+ outflow, the IAA-induced elongation quickly decreases and eventually stops when K+ is absent. A direct correlation between the seed K content and shoot and root elongation may thus appear.

Sodium (Na) is an element associated with potassium concentration, maintaining body fluid homeostasis [63]. The high amount of potassium is a characteristic feature of most plant foods [46,64,65]. A high potassium content in relation to the sodium content is thought to benefit hypertensive individuals [66]. Moreover, the recommended dietary allowance (RDA) for Na and K is 200 and 500 mg, respectively [33], indicating that bottle gourd seeds may be regarded as a rich source of these minerals for the body. Phosphorous (P) and calcium (Ca) are associated with each other and are required for the growth and maintenance of teeth, bones, and muscles [34,67,68]. An adult’s RDA of Ca and P is 810 mg/day [51]. This confirms that bottle-gourd seeds are good sources of phosphorous [69,70,71]. The amount of Mg in bottle gourd seeds is higher than in groundnut, shea nut, cotton seed, and calabash seed [67,72]. Hence, bottle gourd seeds may thus be a rich source of Mg, assuming a total absorption by the body and ignoring the effect of antinutritional factors [33]. Iron (Fe) is the most abundant microelement in crop seeds. A similar trend was observed for the Fe content, with no significant differences (*p* > 0.05). The values in [64,65,73] well support the results. Copper (Cu) is an essential trace element and an integral part of ceruloplasmin (a Cu protein) and cytochrome (an electron carrier). It plays a vital role in energy metabolism [42,74]. High temperatures disrupt Fe homeostasis and inactivate enzymes. The values obtained in the present study seem high compared with the pride of Barbados [66] and calabash seeds [72]. The temperature was directly influenced by zinc (Zn) and manganese (Mn) accumulation in bottle-gourd cultivars. Manganese plays a structural role in the chloroplast membrane and is a cofactor for DNA, RNA, and fatty acid synthesis [75], while Zn is required for growth, tissue repair, and the normal functioning of the immune system [76].

Minerals play a pivotal role in plant nutrition and development [77]. The utilization of mineral components in agriculture, as advocated by Ayurveda, has been a longstanding practice in various traditional farming systems [78,79]. Ayurveda suggests that incorporating mineral-rich preparations into the soil can enrich it with these crucial elements, ensuring that bottle-gourd plants receive adequate mineral nutrition throughout their growth cycle [80]. Ayurvedic practices involve using herbal formulations combined with mineral components, which acts as a carrier for other nutrients, facilitating their absorption by plants [81]. This synergistic approach harnesses the benefits of both herbal and mineral components to promote plant growth and health of bottle gourd.

The results of the germination indices reveal interesting insights into the germination performance of different genotypes and crossing periods in bottle gourd (*Lagenaria siceraria*) cultivation. This finding aligns with previous studies indicating that environmental factors during seed maturation, such as temperature and moisture, can influence the seed quality and germination rates [82]. The Kharif season typically provides more favorable conditions for seed development, potentially contributing to the observed difference in germination percentages. Genotypic variation in germination capacity is well-documented in crop plants [83]. Genetic factors, including seed coat characteristics and embryo vigor, can significantly impact germination rates.

Interestingly, the crossing period also played a crucial role in seed germination. The lowest germination percentage was recorded in seeds produced from crossing period CP1. This result may be attributed to factors such as parental genetic compatibility and seed maturation conditions during that specific crossing period. Subsequent crossing periods, CP2 through CP4, showed a significant increase in the germination percentage, with the maximum germination observed in CP4. This pattern suggests that certain crossing periods may be more conducive to achieving higher germination rates, which could be linked to optimal environmental conditions during seed development [84].

The growth characteristics of hybrid plants derived from the HBGH-35 and G2 parental lines provide additional insights into the impact of the genotype and crossing period on plant development. The significant differences observed in root length, shoot length, root dry mass, and shoot dry mass between HBGH-35 and G2 reaffirm the role of genetic variation in determining plant growth parameters [85]. At CP4, HBGH-35 exhibited the highest root length, suggesting its potential for developing a robust root system, which can be advantageous for nutrient uptake and overall plant stability [86]. Similarly, the variation in shoot length and biomass measurements further emphasizes the genotype-specific responses to crossing periods. The highest shoot length recorded for HBGH-35 at CP1 and CP3 suggests its ability to produce longer and potentially more photosynthetically active shoots under certain conditions [87]. On the other hand, G2 consistently exhibited a lower shoot length and biomass measurements, indicating its comparatively inferior growth characteristics.

Correlation studies provide a better understanding of the association of different traits that may be useful to breeders in selecting cultivars possessing groups of desired agronomical traits [88,89]. Similar results were reported in previous studies [90,91,92,93]. Introducing high-yield cultivars depends upon the agro-climatic conditions and phenological traits and choosing a small number of essential traits with a positive correlation [46,94]. Henceforth, the results showed that all the biochemical components were significantly positively correlated to the minimum temperature and morning and evening relative humidity, suggesting their relationship with temperature and solar radiation.

## 5. Conclusions

In summary, while seeds may contribute modestly to a mature plant’s mineral makeup, they wield a profound impact on the early growth of seedlings. Seeds rich in minerals facilitate rapid seedling establishment, leading to a more vigorous and efficient plant emergence. Our research suggests that dark-colored hybrid HBGH-35 seeds at the fourth week of the crossing period (CP4) hold promise as a source of seedling vigor in nutrient-deficient soils. Bottle gourds’ composition and mineral content varied across different crossing periods, likely due to temperature adaptations. The protein and fiber content increased at CP4, while the fat content decreased. This indicates that the temperature and seedling crossing period strongly influence these constituents. The fourth week of the crossing period during the Kharif season, with a temperature of 31.7 °C, appears favorable for high-quality bottle gourd seed production. Notably, published mineral element values for bottle gourd lines are scarce, making our work a pioneering effort to generate data for future comparisons. Furthermore, with global temperatures on the rise due to climate change, there may be significant implications for crop yields and the nutritional properties of seeds. This underscores the need for continued research to address potential challenges in ensuring food security in changing climate conditions.

## Figures and Tables

**Figure 1 plants-12-03998-f001:**
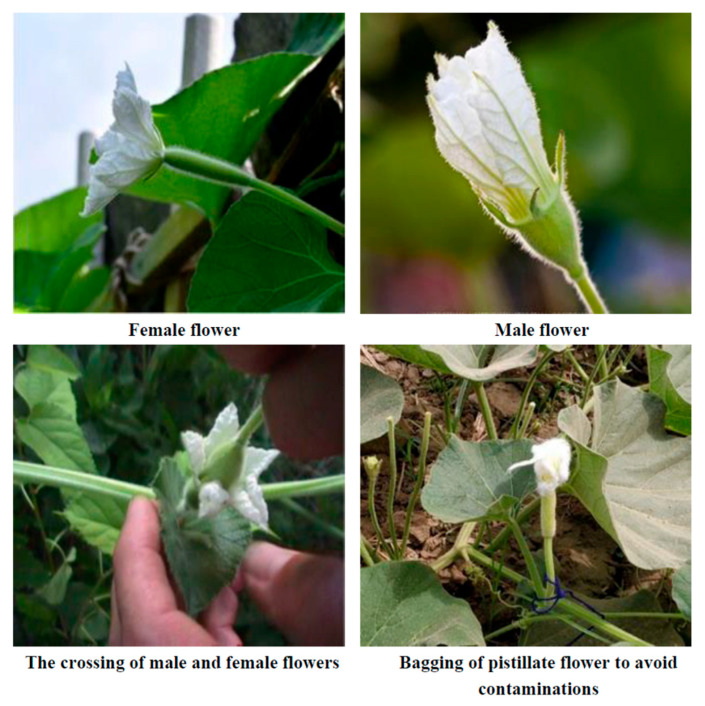
Stepwise emasculation and crossing procedures of bottle gourd male and female flowers.

**Figure 2 plants-12-03998-f002:**
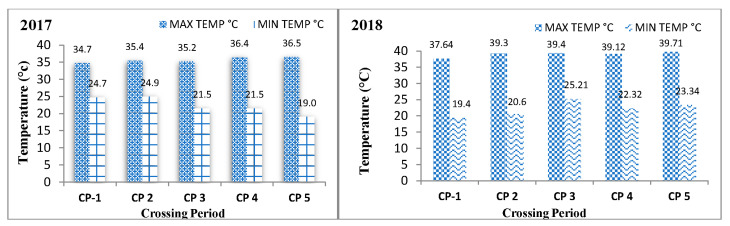
Average maximum and minimum temperature during different crossing periods in bottle gourd during Kharif 2017 and summer 2018.

**Figure 3 plants-12-03998-f003:**
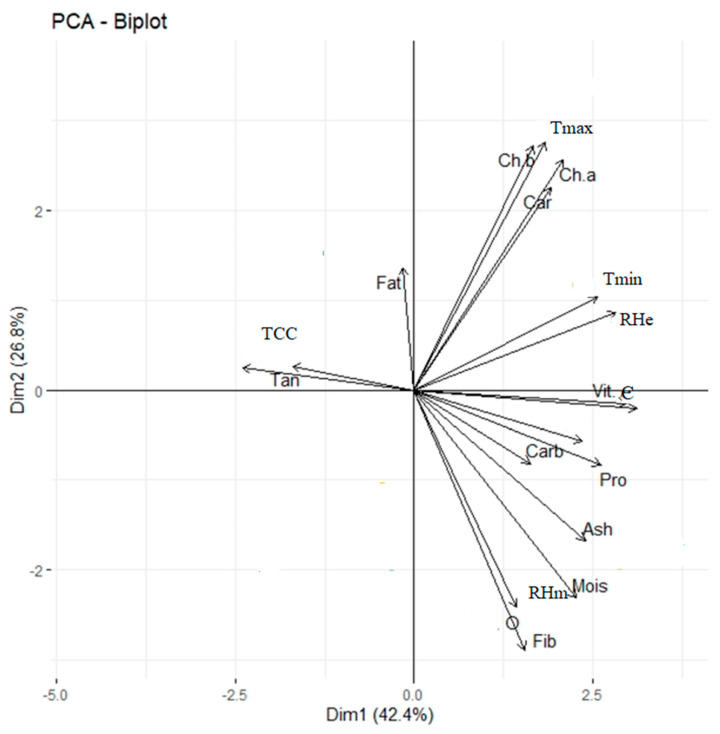
Biplot of principal component analysis (PCA) displaying trait associations under different crossing periods contributing towards PCA based on Cos^2^ Value.

**Figure 4 plants-12-03998-f004:**
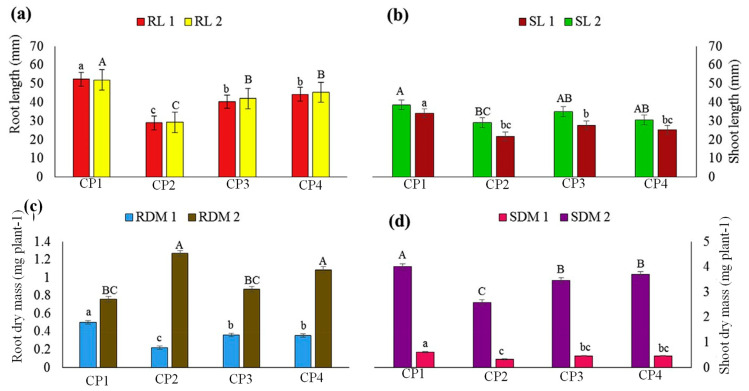
Analysis of bottle gourd parental line growth patterns at CP1 and CP4. (**a**) Root length (RL); (**b**) shoot length (SL); (**c**) root dry mass (RDM), and (**d**) shoot dry mass (SDM). RL 1 (root length of HBGH35), RL 2 (root length of G2 line). Means of significantly different variables were separated using Duncan’s test in GenStat at the 5% level of significance. The averages followed by the same uppercase or lowercase do not differ at 5% level of significance. Lowercase letters compare the values of root length and capital letters compare values throughout the experiment.

**Table 1 plants-12-03998-t001:** Harvesting dates of different crossing periods.

Crossing Periods	Kharif (2017)	Summer Season (2018)
CP-1 (first week of crossing period)	4 September to 10 September 2017	6 April to 22 April 2018
CP-2 (second week of crossing period)	11 September to 17 September 2017	23 April to 29 April 2018
CP-3 (third week of crossing period)	18 September to 24 September 2017	30 April to 6 May 2018
CP-4 (fourth week of crossing period)	25 September to 1 October 2017	7 May to 13 May 2018
CP-5 (fifth week of crossing period)	2–8 October 2017	14 May to 20 May 2018

**Table 2 plants-12-03998-t002:** Meteorological data for 2017 (Kharif season) and 2018 (summer season) of bottle gourd growing seasons.

Months	Kharif Season (2017)							
	T_max_ (°C) *	T_min_ (°C) *	RH_m_ (%) *	RH_e_ (%) *	BRI Sun (Hrs) *	Pan Evap. (mm) *	Rainfall (mm)	Avg. WS * (km/h)
July	34.4	27.0	90.7	70.9	6.5	4.6	2.6	7.3
Aug	34.7	26.3	89.7	69.3	6.3	4.2	3.1	5.6
Sept	34.9	23.5	87.2	49.5	6.8	4.2	1.9	2.9
Oct	35.0	17.2	84.8	28.0	6.6	3.6	0.0	1.9
Nov	27.2	10.8	90.1	39.8	3.4	2.8	0.0	2.0
Dec	23.8	7.3	89.7	34.5	3.7	1.4	0.0	0.7
Mean	31.7	18.7	88.7	48.7	5.5	3.5	1.3	3.4
Summer season (2018)								
	T_max_ (°C) *	T_min_ (°C) *	RH_m_ (%) *	RH_e_ (%) *	BRI Sun (Hrs) *	Pan Evap. * (mm)	Rainfall (mm)	Avg. WS * (km/h)
Feb	28.02	12.80	92.60	48.00	6.48	2.80	0.00	2.84
March	30.92	12.18	82.03	35.87	6.89	3.65	0.00	2.70
April	36.65	19.53	60.33	33.40	7.26	6.41	0.47	5.59
May	40.63	23.73	56.74	28.71	7.05	8.61	0.00	7.05
June	39.58	27.75	71.70	46.87	4.38	8.88	1.96	8.77
July	35.02	26.74	88.00	69.32	4.95	4.76	5.11	5.57
Mean	35.13	20.45	75.23	43.69	6.17	5.85	1.26	5.42

* T_max_: maximum temperature; T_min_: minimum temperature; RH_m_: morning relative humidity; RH_e_: evening relative humidity; BRI: the bright sun; Evap.: evaporation, Avg. WS: average wind speed.

**Table 3 plants-12-03998-t003:** The relative standard deviations, wavelength, and recovery level for elemental evaluation in bottle gourd seeds.

Element	Wavelength (λ)	Recovery Level (%)	RSD (%)
K	590 nm	98.7	5.9
Ca	423 nm	99.8	7.9
Mg	285 nm	97.5	4.8
Zn	214 nm	98.2	6.4
Cu	325 nm	95.1	5.3
Fe	249 nm	98.3	7.5
Mn	280 nm	99.6	7.8

**Table 4 plants-12-03998-t004:** Proximate composition analysis of bottle gourd seeds under two different growing seasons.

Seasons	Moisture (%)		Ash (%)		Crude Fiber (%)	
	Kharif	Summer	Kharif	Summer	Kharif	Summer
Hybrid HBGH-35						
CP-1 *	8.6 ± 0.18 ^e^	8.4 ± 0.16 ^e^	3.5 ± 0.06 ^e^	2.1 ± 0.01 ^e^	20.3 ± 0.56 ^e^	17.4 ± 0.26 ^e^
CP-2	9.4 ± 0.16 ^d^	9.2 ± 0.17 ^d^	3.9 ± 0.01 ^d^	2.7 ± 0.06 ^d^	26.6 ± 0.46 ^d^	20.2 ± 0.49 ^d^
CP-3	10.2 ± 0.18 ^c^	9.8 ± 0.11 ^c^	4.6 ± 0.06 ^b^	3.4 ± 0.09 ^b^	38.5 ± 0.78 ^b^	32.3 ± 0.58 ^b^
CP-4	12.4 ± 0.19 ^b^	11.8 ± 0.10 ^b^	5.8 ± 0.07 ^a^	4.9 ± 0.04 ^a^	48.3 ± 0.69 ^a^	42.6 ± 0.47 ^a^
CP-5	14.3 ± 0.13 ^a^	13.4 ± 0.16 ^a^	4.1 ± 0.06 ^c^	3.1 ± 0.07 ^c^	31.2 ± 0.44 ^c^	27.8 ± 0.65 ^c^
Pusa Naveen (♀)						
CP-1	8.4 ± 0.12 ^e^	8.2 ± 0.23 ^e^	2.5 ± 0.06 ^e^	1.8 ± 0.04 ^d^	20.1 ± 0.36 ^e^	16.7 ± 0.32 ^e^
CP-2	9.5 ± 0.46 ^d^	9.3 ± 0.16 ^d^	2.9 ± 0.08 ^d^	2.1 ± 0.06 ^c^	24.3 ± 0.46 ^d^	20.4 ± 0.22 ^d^
CP-3	10.8 ± 0.18 ^c^	9.8 ± 0.15 ^c^	3.9 ± 0.04 ^b^	3.4 ± 0.09 ^b^	31.2 ± 0.25 ^b^	26.9 ± 0.36 ^b^
CP-4	12.8 ± 0.18 ^b^	11.6 ± 0.19 ^b^	5.2 ± 0.04 ^a^	4.6 ± 0.04 ^a^	37.2 ± 0.65 ^a^	32.6 ± 0.42 ^a^
CP-5	14.6 ± 0.17 ^a^	13.7 ± 0.16 ^a^	3.5 ± 0.03 ^c^	2.1 ± 0.03 ^c^	27.4 ± 0.33 ^c^	23.1 ± 0.87 ^c^
G2-line (♂)						
CP-1	8.7 ± 0.13 ^e^	8.5 ± 0.16 ^e^	1.7 ± 0.02 ^e^	1.3 ± 0.03 ^e^	17.6 ± 0.26 ^e^	13.4 ± 0.36 ^e^
CP-2	9.7 ± 0.14 ^d^	9.5 ± 0.12 ^d^	2.1 ± 0.05 ^d^	1.7 ± 0.06 ^d^	19.6 ± 0.46 ^d^	16.3 ± 0.54 ^d^
CP-3	10.8 ± 0.18 ^c^	10.3 ± 0.14 ^c^	3.1 ± 0.09 ^b^	2.7 ± 0.07 ^b^	28.6 ± 0.26 ^b^	22.3 ± 0.52 ^b^
CP-4	12.7 ± 0.16 ^b^	12.2 ± 0.19 ^b^	4.2 ± 0.08 ^a^	3.8 ± 0.08 ^a^	30.3 ± 0.54 ^a^	27.6 ± 0.41 ^a^
CP-5	14.8 ± 0.12 ^a^	13.7 ± 0.12 ^a^	2.7 ± 0.01 ^c^	2.1 ± 0.09 ^c^	24.2 ± 0.61 ^c^	20.3 ± 0.23 ^c^

^a–e^ Values with different superscripts are significantly different (*p* < 0.05). * CP-1: 4–10 September 2017 and 6–22 April 2018; CP-2: 11–17 September 2017 and 23–29 April 2018; CP-3: 18–24 September 2017 and 30 April–6 May 2018; CP-4: 25 September–1 October 2017 and 7–13 May 2018; CP-5: 2–8 October 2017 and 14–20 May 2018.

**Table 5 plants-12-03998-t005:** Biochemical analysis in bottle gourd seed harvests at two different temperatures.

Seasons	TSS (°Brix)		Crude Protein (%)		Lipid Content (%)	
	Kharif (31.7 °C)	Summer (40 °C)	Kharif (31.7 °C)	Summer (40 °C)	Kharif (31.7 °C)	Summer (40 °C)
Hybrid HBGH-35						
CP-1 *	4.7 ± 0.11 ^e^	4.1 ± 0.11 ^e^	12.7 ± 0.23 ^e^	11.3 ± 0.23 ^e^	21.3 ± 0.32 ^e^	18.6 ± 0.32 ^e^
CP-2	5.6 ± 0.07 ^d^	4.9 ± 0.10 ^d^	13.3 ± 0.32 ^d^	12.5 ± 0.14 ^d^	24.7 ± 0.26 ^d^	20.4 ± 0.21 ^d^
CP-3	7.4 ± 0.10 ^b^	6.4 ± 0.16 ^b^	15.3 ± 0.26 ^b^	14.2 ± 0.19 ^b^	31.7.1 ± 0.21 ^b^	30.5 ± 0.17 ^b^
CP-4	8.2 ± 0.16 ^a^	7.9 ± 0.13 ^a^	19.1 ± 0.21 ^a^	17.6 ± 0.21 ^a^	37.4 ± 0.27 ^a^	35.2 ± 0.16 ^a^
CP-5	6.4 ± 0.12 ^c^	5.6 ± 0.11 ^c^	14.2 ± 0.36 ^c^	13.6 ± 0.16 ^c^	27.6 ± 0.26 ^c^	25.6 ± 0.11 ^c^
Pusa Naveen (♀)						
CP-1	3.4 ± 0.12 ^e^	3.0 ± 0.13 ^e^	9.1 ± 0.26 ^e^	8.6 ± 0.09 ^e^	15.6 ± 0.26 ^e^	12.3 ± 0.23 ^e^
CP-2	4.8 ± 0.14 ^d^	4.1 ± 0.16 ^d^	10.6 ± 0.21 ^d^	9.1 ± 0.16 ^d^	19.6 ± 0.21 ^d^	14.6 ± 0.16 ^d^
CP-3	6.7 ± 0.13 ^b^	5.7 ± 0.15 ^b^	14.3 ± 0.19 ^b^	11.3 ± 0.15 ^b^	25.1 ± 0.16 ^b^	22.1 ± 0.17 ^b^
CP-4	7.6 ± 0.16 ^a^	6.2 ± 0.14 ^a^	16.6 ± 0.16 ^a^	14.3 ± 0.21 ^a^	32.3 ± 0.36 ^a^	29.6 ± 0.35 ^a^
CP-5	5.7 ± 0.12 ^c^	5.1 ± 0.13 ^c^	12.3 ± 0.21 ^c^	10.6 ± 0.36 ^c^	22.6 ± 0.23 ^c^	19.6 ± 0.26 ^c^
G2-line (♂)						
CP-1	3.4 ± 0.16 ^e^	2.9 ± 0.10 ^e^	8.2 ± 0.11 ^e^	6.2 ± 0.06 ^e^	11.6 ± 0.18 ^e^	9.1 ± 0.11 ^e^
CP-2	4.1 ± 0.22 ^d^	3.6 ± 0.09 ^d^	8.6 ± 0.13 ^d^	7.6 ± 0.16 ^d^	14.6 ± 0.21 ^d^	11.0 ± 0.26 ^d^
CP-3	5.4 ± 0.19 ^b^	4.9 ± 0.16 ^b^	10.3 ± 0.23 ^b^	9.6 ± 0.19 ^b^	23.6 ± 0.26 ^b^	20.6 ± 0.24 ^b^
CP-4	6.5 ± 0.23 ^a^	6.0 ± 0.14 ^a^	13.3 ± 0.20 ^a^	11.2 ± 0.21 ^a^	27.6 ± 0.11 ^a^	24.3 ± 0.32 ^a^
CP-5	4.7 ± 0.11 ^c^	4.1 ± 0.17 ^c^	9.6 ± 0.16 ^c^	8.6 ± 0.19 ^c^	20.3 ± 0.19 ^c^	17.6 ± 0.17 ^c^

^a–e^ Values with different superscripts are significantly different (*p* < 0.05). * CP-1: 4–10 September 2017 and 6–22 April 2018; CP-2: 11–17 September 2017 and 23–29 April 2018; CP-3: 18–24 September 2017 and 30 April–6 May 2018; CP-4: 25 September–1 October 2017 and 7–13 May 2018; CP-5: 2–8 October 2017 and 14–20 May 2018; CP: crossing period.

**Table 6 plants-12-03998-t006:** Total carbohydrates and antioxidants in bottle gourd seeds under different growing seasons.

Seasons	TCC (%)		Vitamin C (mg/100 g)		Energy Values (kcal/100 g)	
	Kharif (31.7 °C)	Summer (40 °C)	Kharif (31.7 °C)	Summer (40 °C)	Kharif (31.7 °C)	Summer (40 °C)
Hybrid HBGH-35						
CP-1 *	10.3 ± 0.23 ^e^	9.4 ± 0.12 ^e^	13.1 ± 0.23 ^e^	12.4 ± 0.16 ^e^	196.2 ± 23.6 ^e^	154.3 ± 26.6 ^e^
CP-2	12.6 ± 0.15 ^d^	10.3 ± 0.16 ^d^	14.2 ± 0.32 ^d^	13.7 ± 0.23 ^d^	247.3 ± 23.6 ^d^	202.5 ± 41.2 ^d^
CP-3	19.5 ± 0.16 ^b^	17.9 ± 0.26 ^b^	15.8 ± 0.24 ^b^	15.2 ± 0.26 ^b^	385.6 ± 35.6 ^b^	320.1 ± 52.6 ^b^
CP-4	22.1 ± 0.25 ^a^	20.3 ± 0.41 ^a^	16.5 ± 0.16 ^a^	16.0 ± 0.16 ^a^	456.3 ± 37.6 ^a^	412.3 ± 38.6 ^a^
CP-5	15.6 ± 0.14 ^c^	13.6 ± 0.16 ^c^	14.9 ± 0.21 ^c^	14.2 ± 0.32 ^c^	310.3 ± 41.3 ^c^	287.3 ± 23.6 ^c^
Pusa Naveen (♀)						
CP-1	9.5 ± 0.12 ^e^	8.4 ± 0.10 ^e^	8.7 ± 0.23 ^e^	7.8 ± 0.12 ^e^	142.3 ± 32.6 ^e^	121.3 ± 26.6 ^e^
CP-2	10.6 ± 0.23 ^d^	9.2 ± 0.09 ^d^	9.5 ± 0.16 ^d^	8.0 ± 0.16 ^d^	185.6 ± 42.6 ^d^	154.2 ± 22.6 ^d^
CP-3	17.6 ± 0.19 ^b^	13.8 ± 0.16 ^b^	11.6 ± 0.18 ^b^	9.2 ± 0.21 ^b^	274.6 ± 23.3 ^b^	254.1 ± 28.6 ^b^
CP-4	19.2 ± 0.21 ^a^	16.5 ± 0.35 ^a^	14.6 ± 0.19 ^a^	12.1 ± 0.19 ^a^	385.6 ± 56.4 ^a^	365.6 ± 38.4 ^a^
CP-5	13.2 ± 0.16 ^c^	11.7 ± 0.16 ^c^	10.3 ± 0.21 ^c^	8.6 ± 0.16 ^c^	212.6 ± 23.6 ^c^	174.8 ± 21.3 ^c^
G2-line (♂)						
CP-1	8.9 ± 0.23 ^e^	8.1 ± 0.12 ^e^	7.4 ± 0.13 ^e^	6.8 ± 0.10 ^e^	132.6 ± 23.6 ^e^	102.3 ± 36.2 ^e^
CP-2	9.6 ± 0.16 ^d^	8.7 ± 0.16 ^d^	8.6 ± 0.16 ^d^	7.5 ± 0.26 ^d^	142.3 ± 32.6 ^d^	112.5 ± 24.6 ^d^
CP-3	13.2 ± 0.19 ^b^	10.6 ± 0.25 ^b^	10.6 ± 0.23 ^b^	8.7 ± 0.21 ^b^	232.3 ± 21.4 ^b^	206.3 ± 35.2 ^b^
CP-4	15.5 ± 0.11 ^a^	12.3 ± 0.16 ^a^	12.6 ± 0.26 ^a^	10.6 ± 0.16 ^a^	287.6 ± 23.6 ^a^	255.2 ± 29.6 ^a^
CP-5	11.6 ± 0.17 ^c^	9.4 ± 0.18 ^c^	9.4 ± 0.27 ^c^	7.9 ± 0.06 ^c^	184.2 ± 21.2 ^c^	149.6 ± 20.4 ^c^

^a–e^ Values with different superscripts are significantly different (*p* < 0.05). * CP-1: 4–10 September 2017 and 6–22 April 2018; CP-2: 11–17 September 2017 and 23–29 April 2018; CP-3: 18–24 September 2017 and 30 April–6 May 2018; CP-4: 25 September–1 October 2017 and 7–13 May 2018; CP-5: 2–8 October 2017 and 14–20 May 2018; CP: crossing period.

**Table 7 plants-12-03998-t007:** Macro-nutrients analysis of bottle gourd parental lines by EDX analysis.

	Phosphorus (%)			Potassium (%)			Magnesium (%)		
Crossing Periods	Hybrid	Pusa Naveen	G2	Hybrid	Pusa Naveen	G2	Hybrid	Pusa Naveen	G2
CP1	53.10 ^b^	13.24 ^d^	2.30 ^b^	28.78 ^a^	7.46 ^a^	0.84 ^d^	16.12 ^b^	4.03 ^b^	1.07 ^b^
CP2	52.73 ^d^	14.30 ^a^	2.29 ^a^	29.38 ^b^	6.76 ^c^	0.85 ^c^	15.73 ^c^	3.93 ^c^	1.05 ^c^
CP3	53.03 ^c^	13.60 ^b^	2.29 ^a^	29.22 ^c^	6.51 ^d^	0.87 ^b^	15.70 ^d^	3.93 ^c^	1.05 ^c^
CP4	53.13 ^a^	13.50 ^c^	2.3 ^b^	29.11 ^d^	6.92 ^b^	1.12 ^a^	16.35 ^a^	4.09 ^a^	1.09 ^a^
Mean	52.1	14.01	3.29	30.12	7.11	0.92	16.01	4.2	1.06
(*p* < 0.05)	0.92	0.89	0.95	0.93	0.68	0.20	0.42	0.41	0.39
l.s.d	1.48	2.65	0.06	2.14	1.81	0.32	1.00	0.25	0.06
cv%	1.50	13.30	1.50	2.90	14.00	17.60	2.30	2.30	2.30

^a–d^ Values with different superscripts are significantly different (*p* < 0.05).

**Table 8 plants-12-03998-t008:** The micro-nutrients analysis of bottle gourd parental lines using EDX analysis.

Region	Parental Lines	Ca (%)	Zn (%)	Cu (%)	Mn (%)
Embryo	CP1	1.01 ^a^	1.52 ^a^	0.373 ^a^	0.29 ^a^
	CP2	0.45 ^d^	0.24 ^d^	0.10 ^d^	0.06 ^d^
	CP3	0.74 ^c^	0.45 ^c^	0.30 ^c^	0.10 ^c^
	CP4	0.82 ^b^	0.80 ^b^	0.34 ^b^	0.16 ^b^
	l.s.d (P¼0.05)	0.16	0.16	0.01	0.041
Cotyledon	CP1	1.060 ^a^	0.810 ^a^	0.19 ^b^	0.077 ^a^
	CP2	0.507 ^c^	0.131 ^d^	0.08 ^d^	0.075 ^c^
	CP3	0.793 ^b^	0.245 ^c^	0.124 ^c^	0.032 ^ab^
	CP4	0.873 ^b^	0.426 ^b^	0.218 ^a^	0.054 ^a^
	l.s.d (P¼0.05)	0.085	0.091	0.014	0.027
Seed Coat	CP1	2.02 ^a^	0.603 ^a^	0.171 ^a^	0.055 ^a^
	CP2	0.74 ^d^	0.094 ^d^	0.99 ^c^	0.01 ^c^
	CP3	0.95 ^c^	0.183 ^c^	0.016 ^d^	0.041 ^b^
	CP4	1.301 ^b^	0.320 ^b^	1.59 ^b^	0.038 ^b^
	l.s.d (P¼0.05)	0.085	0.065	0.019	0.027

Means were separated using Duncan’s test. Means with the same letters within a column are not significantly different and means with different letters within a column are significantly different.

**Table 9 plants-12-03998-t009:** Mineral composition of hybrid bottle gourd under different Kharif seasons.

Season	Kharif Season (2017–2018)				
Crossing Periods	CP-1	CP-2	CP-3	CP-4	CP-5
Hybrid HBGH-35					
Sodium	1042.6 ± 1.2 ^a^	1154.6 ± 1.6 ^b^	131.74.3 ± 1.6 ^d^	1412.2 ± 1.5 ^e^	1265.3 ± 1.3 ^c^
Potassium	2054.2 ± 4.8 ^a^	2145.3 ± 5.3 ^b^	2654.2 ± 2.3 ^d^	3015.2 ± 6.6 ^e^	2436.0 ± 3.2 ^c^
Calcium	1.89 ± 0.23 ^e^	2.69 ± 0.16 ^d^	3.65 ± 0.18 ^b^	3.80 ± 0.16 ^a^	3.23 ± 0.15 ^c^
Phosphorus	976.6 ± 4.2 ^a^	1065.3 ± 2.3 ^b^	1224.3 ± 6.3 ^d^	1250.3 ± 2.3 ^e^	1165.3 ± 4.2 ^c^
Magnesium	300.6 ± 2.2 ^a^	385.6 ± 2.3 ^b^	541.3 ± 4.5 ^d^	570.6 ± 6.3 ^e^	475.2 ± 5.6 ^c^
Iron	31.2 ± 1.2 ^e^	35.3 ± 1.6 ^d^	54.2 ± 1.5 ^b^	65.3 ± 2.6 ^a^	48.2 ± 2.2 ^c^
Copper	9.6 ± 0.23 ^e^	11.4 ± 0.26 ^d^	13.2 ± 0.45 ^b^	15.2 ± 0.56 ^a^	12.4 ± 0.36 ^c^
Zinc	0.10 ± 0.03 ^a^	0.11 ± 0.09 ^b^	0.16 ± 0.02 ^d^	0.20 ± 0.11 ^e^	0.13 ± 0.1 ^c^
Manganese	16.5 ± 0.32 ^e^	21.6 ± 0.16 ^d^	23.3 ± 0.12 ^b^	26.4 ± 0.16 ^a^	20.3 ± 0.18 ^c^
Pusa Naveen (♀)					
Sodium	982.6 ± 1.6 ^a^	1089.6 ± 1.3 ^b^	1289.6 ± 1.2 ^d^	1324.6 ± 1.1 ^e^	1212.4 ± 1.4 ^c^
Potassium	1965.3 ± 4.5 ^a^	2084.3 ± 4.6 ^b^	2464.2 ± 3.2 ^d^	2865.3 ± 2.6 ^e^	2246.2 ± 2.9 ^c^
Calcium	1.6 ± 0.36 ^e^	2.3 ± 0.36 ^d^	3.2 ± 0.45 ^b^	3.5 ± 0.55 ^a^	2.7 ± 0.16 ^c^
Phosphorus	932.3 ± 4.2 ^a^	985.6 ± 3.2 ^b^	1056.3 ± 2.1 ^d^	1145.3 ± 4.3 ^e^	1041.3 ± 4.6 ^c^
Magnesium	265.3 ± 4.2 ^a^	289.6 ± 3.6 ^b^	463.2 ± 3.1 ^d^	485.3 ± 4.2 ^e^	421.6 ± 3.6 ^c^
Iron	21.6 ± 1.6 ^e^	27.6 ± 2.3 ^d^	47.2 ± 2.6 ^b^	51.2 ± 2.1 ^a^	36.4 ± 1.6 ^c^
Copper	8.6 ± 0.10 ^e^	9.5 ± 0.09 ^d^	10.6 ± 10.16 ^b^	12.5 ± 0.18 ^a^	10.1 ± 0.16 ^c^
Zinc	0.07 ± 0.03 ^a^	0.09 ± 0.02 ^b^	0.14 ± 0.02 ^d^	0.18 ± 0.04 ^e^	0.11 ± 0.09 ^c^
Manganese	13.6 ± 0.10 ^e^	17.4 ± 0.11 ^d^	20.6 ± 0.16 ^b^	23.1 ± 0.17 ^a^	18.6 ± 0.14 ^c^
G2-line (♂)					
Sodium	921.3 ± 1.2 ^a^	1021.3 ± 1.3 ^b^	1242.2 ± 1.5 ^d^	1265.3 ± 1.2 ^e^	1176.3 ± 2.3 ^c^
Potassium	1945.2 ± 5.2 ^a^	2021.3 ± 4.2 ^b^	2364.3 ± 3.1 ^d^	2445.3 ± 3.6 ^e^	2145.3 ± 2.9 ^c^
Calcium	1.3 ± 0.23 ^e^	1.8 ± 0.21 ^d^	2.6 ± 0.12 ^b^	3.2 ± 0.16 ^a^	2.2 ± 0.18 ^c^
Phosphorus	881.3 ± 4.6 ^a^	945.2 ± 3.2 ^b^	1022.3 ± 2.3 ^d^	1121.3 ± 4.3 ^e^	1005.3 ± 5.6 ^c^
Magnesium	232.6 ± 4.2 ^a^	231.7.3 ± 3.2 ^b^	411.6 ± 5.1 ^d^	449.6 ± 2.6 ^e^	359.6 ± 3.6 ^c^
Iron	19.6 ± 2.1 ^e^	21.3 ± 1.6 ^d^	37.4 ± 1.4 ^b^	44.2 ± 1.2 ^a^	31.6 ± 1.3 ^c^
Copper	7.2 ± 0.09 ^e^	8.3 ± 0.10 ^d^	9.1 ± 0.12 ^b^	10.4 ± 0.16 ^a^	9.4 ± 0.11 ^c^
Zinc	0.05 ± 0.01 ^a^	0.06 ± 0.02 ^b^	0.09 ± 0.06 ^d^	0.15 ± 0.04 ^e^	0.08 ± 0.04 ^c^
Manganese	10.6 ± 0.13 ^e^	15.9 ± 0.10 ^d^	18.6 ± 0.11 ^b^	20.3 ± 0.13 ^a^	14.6 ± 0.06 ^c^

^a–e^ Values with different superscripts are significantly different (*p* < 0.05); CP-1: 4–10 September 2017 and 6–22 April 2018; CP-2: 11–17 September 2017 and 23–29 April 2018; CP-3: 18–24 September 2017 and 30 April–6 May 2018; CP-4: 25 September-1 October 2017 and 7–13 May 2018; CP-5: 2–8 October 2017 and 14–20 May 2018; CP: crossing period.

**Table 12 plants-12-03998-t012:** The germination percent of bottle gourd lines under different crossing periods.

Season (S)								
Genotype → Crossing Period ↓	Kharif 2017				Summer 2018			
	HBGH-35	Pusa Naveen (♀)	G-_2_ (♂)	Mean	HBGH-35	Pusa Naveen (♀)	G-_2_ (♂)	Mean
CP_1_	96.0 ^d^ (78.49)	95.0 ^e^ (77.05)	94.67 ^d^ (76.63)	95.22 ^e^ (77.39)	94.0 ^d^ (75.79)	93.0 ^d^ (74.63)	92.67 ^d^ (74.27)	93.22 (74.90)
CP_2_	96.33 ^b^ (78.95)	95.33 ^d^ (77.51)	95.0 ^c^ (77.09)	95.56 ^d^ (77.85)	94.33 ^c^ (76.21)	94.0 ^c^ (75.79)	93.0 ^c^ (74.63)	93.78 (75.54)
CP_3_	97.0 ^c^ (80.09)	96. ^c^ (78.43)	95.33 ^b^ (77.51)	96.11 ^b^ (78.68)	94.67 ^b^ (76.63)	94.33 ^b^ (76.21)	93.67 ^b^ (75.40)	94.22 (76.08)
CP_4_	97.67 ^a^ (81.22)	96.33 ^a^ (78.95)	96.0 ^a^ (78.43)	96.67 ^a^ (79.54)	95.33 ^a^ (77.51)	95.67 ^a^ (77.97)	94.67 ^a^ (76.63)	95.22 (77.37)
CP_5_	96.0 ^d^ (78.43)	95.67 ^b^ (77.97)	95.33 ^b^ (77.51)	95.67 ^c^ (77.97)	94.67 ^b^ (76.63)	94.0 ^c^ (75.79)	93.67 ^b^ (75.40)	94.11 (75.94)
Mean	96.60 (79.44)	95.68 (77.98)	95.27 (77.43)	95.54 (78.28)	94.60 (76.55)	94.20 (76.08)	93.53 (75.27)	

^a–e^ Values with different superscripts are significantly different (*p* < 0.05); Values in the bracket represent arcsine transformation.

## Data Availability

Data are contained within the article.

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
