# Peer review of "Investigating Mineral Accumulation and Seed Vigor Potential in Bottle Gourd (*Lagenaria siceraria*) through Crossbreeding Timing"

_plants, 2023, doi:10.3390/plants12233998_

Round 1

Reviewer 1 Report

Comments and Suggestions for Authors

1. Add "seed" on the title "Investigating Plant Mineral Accumulation and Seed Vigour Potential  in Bottle Gourd (Lagenaria siceraria) through Crossbreeding  Timing"

2. Give more information on "kharif season"

3. Line 30: what is F4 crossing period? while you have only C1-C5 crossing period.

4. Figure 2: please give the same Y axis value of both graph (2017 and 2018 graph).

5. Line 244-261: some sentences are contradictive, especially on moisture content and seed quality.

6. some data are misinterpreted

7. lack of statistical analyses on differences among varieties/genotypes.

8. some discussions are contradictive

9. some discussions include Ayurveda use of mineral components, but lack of discussion on their role in seed germination and seedling  development.

10. please check the review comment on the manuscript.

Comments on the Quality of English Language

English is Ok, but much better to write shorter and more effective English.

Author Response

Pointwise answers to reviewer’s comments

Minor editing of English language required: English language and style are needful done as highlighted and minor spell check is also needful done.

S.No

Comment

Reply

Reviewer #1:

Are all the cited references relevant to the research? Can be improved

The citations have been improved.

Are the methods adequately described?

Must be improved

The methods have been corrected as per the suggestions.

Are the results clearly presented? Can be improved

The results have been improved as suggested.

Are the conclusions supported by the results? Can be improved

Rewrote the conclusion to support the findings of the present study.

Kindly refer to line no. 835-850.

1

Add "seed" on the title "Investigating Plant Mineral Accumulation and Seed Vigour Potential  in Bottle Gourd (Lagenaria siceraria) through Crossbreeding  Timing"

Added in the title.

Kindly refer to line no. 2.

2

Give more information on "kharif season"

Added the information on "kharif season".

Kindly refer to line no. 46-47.

3

 Line 30: what is F4 crossing period? while you have only C1-C5 crossing period.

Replaced the F4 with CP4.

Kindly refer to line no. 56.

4

Figure 2: please give the same Y axis value of both graph (2017 and 2018 graph).

Added the same values on Y axis of both graphs.

Kindly refer to line no. 253-255.

5

Line 244-261: some sentences are contradictive, especially on moisture content and seed quality.

Rectified the statements.

Kindly refer to line no. 690-694.

6

some data are misinterpreted

Corrected the data.

7

lack of statistical analyses on differences among varieties/genotypes.

Needful done.

8

some discussions are contradictive

Improved the text.

Kindly refer to line no. 684-813.

9

some discussions include Ayurveda use of mineral components, but lack of discussion on their role in seed germination and seedling  development.

Added.

Kindly refer to line no. 767-775.

10

please check the review comment on the manuscript.

All the comments have been incorporated in the manuscript.

Reviewer 2 Report

Comments and Suggestions for Authors

Thank you for considering me to revise the manuscript entitled “Investigating plant mineral accumulation and vigour potential in bottle gourd (Lagenaria siceraria) through crossbreeding timing”. The manuscript exhibited the impact of agro-climatic conditions on the quality and biochemical components of two bottle gourd parental lines and one hybrid. I suggest accepting the manuscript following major revisions.

My suggestions

The manuscript needs major English editing. 

Line 19: Scientific names should be in italic (Lagenaria siceraria) and also all scientific names throughout the manuscript. 

Line 30: “F4 crossing period” “F4” should be replaced by “CP4”. The same abbreviations should be unified throughout the manuscripts. 

The introduction needs major improvement. Lines 50-60 should be presented earlier at the beginning of the introduction. The other paragraphs need to be rearranged to present the introduction of studied topic smoothly. The knowledge gap, rationale, and objectives need more clarification.

The materials and methods section needs to be carefully revised and improved.  

Line 116: “The flowers (male and female) were emasculated” how did you collect pollen from emasculated male flowers.

The growing conditions of parental lines and agricultural practices should be presented.

 The results section is poorly written, more efforts are needed to present the text in better status. The difference among assessed crossing periods needs more clarification in all evaluated parameters.
The significance letters are wrongly presented with non-corresponding values. For example in Table 4 in the column of kharif for moisture %, the highest value of the crossing period is 14.3 followed by 12.4, 10.2, 9.4, 8.6. So, they should be accompanied by a, b, c, d and e in the same order. But, the values are presented as 12.4 a, 10.2 b, 14.3 c, 9.4 d and 8.6 e. The same statistical mistake is in the other columns.

In some columns in Table 4, the authors ordered the significance letters from highest to lowest values. While in other columns the authors did the opposite and ordered the significance letters from lowest to highest as Crude fiber, 20.3 a, 26.6 b, 31.2 c, 38.5  d and 48.3 e. Also in Table 5 kharif (31.7°C) of TSS (°Brix), the averages were ordered as 4.7 a, 5.6 b, 6.4 c, 7.4 d, and 8.2 e. But in other columns as kharif (31.7°C) of crude protein ordered from the highest to the lowest 19.1 a, 15.3 b, 14.2 c, 13.3 d, 12.7 e. The presentation of obtained data should be unified throughout the manuscript.

There are no significance letters in Table 7 and 11

Table 9 is poorly presented, it could be divided into separate tables and improved. 

The association among evaluated characteristics and assessed crossing periods could be presented better in a biplot of principal components and heatmap with clustering.

More details are needed in the caption for uppercase and lowercase letters in Figure  3.

The results are poorly discussed,  the discussion should be separated and improved.

Line 549 “TRITICUM DURUM” is written a wrong way, it should be “Triticum durum”. Al scientific names in the text and reference list should be revised.

Line 563 “International Journal of Agricultural Sciences”  The journal in complete name while the others are abbreviated. The reference style should be carefully revised and unified according to Plants style.

Comments on the Quality of English Language

The manuscript needs major English editing. 

Author Response

Pointwise answers to reviewer’s comments

Moderate editing of English language required: English language and style are needful done as highlighted and minor spell check is also needful done.

S.No

Comment

Reply

Reviewer #2:

1

Line 19: Scientific names should be in italic (Lagenaria siceraria) and also all scientific names throughout the manuscript. 

Needful done.

Kindly refer to line no. 43,62.

2

Line 30: “F4 crossing period” “F4” should be replaced by “CP4”. The same abbreviations should be unified throughout the manuscripts. 

Replaced the word in the text.

Kindly refer to line no. 56.

3

The introduction needs major improvement. Lines 50-60 should be presented earlier at the beginning of the introduction. The other paragraphs need to be rearranged to present the introduction of studied topic smoothly. The knowledge gap, rationale, and objectives need more clarification.

The introduction section is rewritten keeping in mind more clarification about the objectives.

Kindly refer to line no. 67-179.

4

The materials and methods section needs to be carefully revised and improved.  

Carefully revised the materials and methods.

Kindly refer to line no. 182-342.

5

Line 116: “The flowers (male and female) were emasculated” how did you collect pollen from emasculated male flowers.

At the start of the crossing period in hybrid and parental lines seed production, the male flower was chosen in the parental lines and bagged with paper bags in the previous evening and pinched off in the next morning and kept carefully in a glass container. After bagging, waited for one week and observed female flower ovary growth. The increased size of the ovary confirmed the successful crossing after that it was tagged using different color threads as shown in the picture.

Kindly refer to line no. 209-232.

6

The growing conditions of parental lines and agricultural practices should be presented.

Added the text.

Kindly refer to line no. 87-190, 203-2081.  

7

The results section is poorly written, more efforts are needed to present the text in better status. The difference among assessed crossing periods needs more clarification in all evaluated parameters.

The results are revised and rectified the sentences.

Kindly refer to line no. 343-684.

8

The significance letters are wrongly presented with non-corresponding values. For example in Table 4 in the column of kharif for moisture %, the highest value of the crossing period is 14.3 followed by 12.4, 10.2, 9.4, 8.6. So, they should be accompanied by a, b, c, d and e in the same order. But, the values are presented as 12.4 a, 10.2 b, 14.3 c, 9.4 d and 8.6 e. The same statistical mistake is in the other columns.

The statistical mistake has been rectified as suggested in Table 4.

Kindly refer to line no. 397.

9

In some columns in Table 4, the authors ordered the significance letters from highest to lowest values. While in other columns the authors did the opposite and ordered the significance letters from lowest to highest as Crude fiber, 20.3 a, 26.6 b, 31.2 c, 38.5  d and 48.3 e. Also in Table 5 kharif (31.7°C) of TSS (°Brix), the averages were ordered as 4.7 a, 5.6 b, 6.4 c, 7.4 d, and 8.2 e. But in other columns as kharif (31.7°C) of crude protein ordered from the highest to the lowest 19.1 a, 15.3 b, 14.2 c, 13.3 d, 12.7 e. The presentation of obtained data should be unified throughout the manuscript.

The data has been rectified in all tables.

10

There are no significance letters in Table 7 and 11

Significant letters have been added in Table 7 and 11.

Kindly refer to line no. 505 and 515.

11

Table 9 is poorly presented, it could be divided into separate tables and improved.

Corrected the Table 9.

Kindly refer to line no.  575 and 585.

12

The association among evaluated characteristics and assessed crossing periods could be presented better in a biplot of principal components and heatmap with clustering.

Added the biplot.

Kindly refer to line no.  647.

13

More details are needed in the caption for uppercase and lowercase letters in Figure  3.

Added the caption.

Kindly refer to line no. 676-683.

14

The results are poorly discussed,  the discussion should be separated and improved.

Discussion is improved and separated from results.

Kindly refer to line no. 343-683 and 684-813.

15

Line 549 “TRITICUM DURUM” is written a wrong way, it should be “Triticum durum”. Al scientific names in the text and reference list should be revised.

Corrected the name as suggested.

Kindly refer to line no.  1082.

16

Line 563 “International Journal of Agricultural Sciences”  The journal in complete name while the others are abbreviated. The reference style should be carefully revised and unified according to Plants style.

The reference style has been updated as per the journal's pattern.

Round 2

Reviewer 2 Report

Comments and Suggestions for Authors

The Authors have addressed major previous comments.
Minor aspects still should be addressed.

Line 199: More details are needed for the male flower. For example, “The male flowers were identified in the parental lines at the beginning of the crossing period and bagged in the previous evening and pinched off in the next morning and kept carefully in a glass container. Then the female flowers “The female flowers were emasculated one day before opening, and they were wrapped in a thin layer of cotton.”

Figures 4 a-d. There are two types of significant letters; uppercase and lowercase, the difference between the two types should be clarified in the caption.

The reference list still needs careful revision. Some journals are abbreviated (as Front Plant Sci line 867 ) while others in complete names (as Current Issues in Molecular Biology in lines 855). Scientific names should be in italics as in lines 872, 876, 879.

Comments on the Quality of English Language

Minor editing of English language required

Author Response

Pointwise answers to reviewer’s comments

Minor editing of English language required: English language and style are needful done as highlighted and minor spell check is also needful done.

S.No

Comment

Reply

Reviewer #2:

1

Line 199: More details are needed for the male flower. For example, “The male flowers were identified in the parental lines at the beginning of the crossing period and bagged in the previous evening and pinched off in the next morning and kept carefully in a glass container. Then the female flowers “The female flowers were emasculated one day before opening, and they were wrapped in a thin layer of cotton.”

Monoecious bottle gourds (Lagenaria siceraria) bear male and female flowers. Key inferior ovary variations between male and female flowers:

Male Flowers: The male flower has a long, slender peduncle and one pollen-producing stamen. Male flowers lack or have a primitive ovary. Male bottle-gourd flowers lack or have a primitive ovary.

Female flowers have a unique ovary at the base that will produce fruit if fertilized. The bulbous ovary is at the floral center. Well-developed female flowers have ovules in their ovaries. A functioning ovary is essential for female flowers.

Bottle gourds and other monoecious plants generate more male flowers compared to female flowers. This may be an adaptive technique to ensure pollination with enough pollen. Environmental conditions and plant health affect flower output and ratio.

To determine flower sex and optimize pollination, growers watch the flowering pattern. If natural pollination is insufficient, farmers may hand-pollinate to increase fruit set.

Kindly refer to line no. 203-215.

2

Figures 4 a-d. There are two types of significant letters; uppercase and lowercase, the difference between the two types should be clarified in the caption.

The averages followed by the same uppercase or lowercase do not differ at 5% level of significance. Lowercase letters compare the values of root length and capital letters compare values throughout the experiment.

Kindly refer to line no. 685-687.

3

The reference list still needs careful revision. Some journals are abbreviated (as Front Plant Sci line 867 ) while others in complete names (as Current Issues in Molecular Biology in lines 855). Scientific names should be in italics as in lines 872, 876, 879.

Rectified the reference list as per the journals format.
